# The Value of Bone Marrow Assessment by FDG PET/CT, Biopsy and Aspirate in the Upfront Evaluation of Mantle Cell Lymphoma: A Nationwide Cohort Study

**DOI:** 10.3390/cancers16244189

**Published:** 2024-12-16

**Authors:** Isabel Ródenas Quiñonero, Javier Marco-Ayala, Tzu-Hua Chen-Liang, Fátima de la Cruz-Vicente, Tycho Baumann, José-Tomás Navarro, Alejandro Martín García-Sancho, Taida Martin-Santos, Javier López-Jiménez, Rafael Andreu, Esther Parra-Virto, Andrea Usas, David Alonso, Marta Fernández-González, Pablo Palomo Rumschisky, Laura Frutos, José Luis Navarro, Rosa María Alvarez-Perez, Pilar Sarandeses, Montserrat Cortes, Pilar Tamayo, Jon Uña, Alberto Martínez-Lorca, Cristina Ruiz, María Luisa Lozano, Francisco José Ortuño

**Affiliations:** 1Servicio de Hematología, Hospital José María Morales Meseguer, IMIB-Pascual Parrilla, Centro de Investigación Biomédica en Red. Enfermedades Raras (CIBERER), 30008 Murcia, Spain; isaa.rq3@gmail.com (I.R.Q.); tzuchen82@gmail.com (T.-H.C.-L.); mllozano@um.es (M.L.L.); fortunog@sehh.es (F.J.O.); 2Department of Medicine, University of Murcia, 30100 Murcia, Spain; 3Servicio de Hematología, Hospital Universitario Virgen del Rocío, Instituto de Biomedicina de Sevilla (IBIS)/Consejo Superior de Investigaciones Científicas (CSIC)/Universidad de Sevilla, 41013 Sevilla, Spain; fatimadelacruzv@gmail.com; 4Servicio de Hematología, Hospital 12 de Octubre, 28041 Madrid, Spain; tycho.baumann@gmail.com (T.B.); esther.parra@salud.madrid.org (E.P.-V.); 5Lymphoid Neoplasms Group, Josep Carreras Leukaemia Research Institute (IJC), Department of Hematology, ICO-Hospital Germans Trias i Pujol Hospital, 08916 Badalona, Spain; jnavarro@iconcologia.net (J.-T.N.); ausas@carrerasresearch.org (A.U.); 6Servicio de Hematología, Hospital Universitario Salamanca, Instituto de Investigación Biomédica de Salamanca (IBSAL), Centro de Investigación Biomédica en Red. Cáncer (CIBERONC), University of Salamanca, 37008 Salamanca, Spain; amartingar@usal.es (A.M.G.-S.); dalonsoca@saludcastillayleon.es (D.A.); 7Servicio de Hematología, Hospital Universitario de Canarias, 38320 La Laguna, Spain; taidamar@hotmail.com (T.M.-S.); mfernandez7976@gmail.com (M.F.-G.); 8Servicio de Hematología, Hospital Ramón y Cajal, 28034 Madrid, Spain; jljimenez@salud.madrid.org (J.L.-J.); pablopalomorumschisky@gmail.com (P.P.R.); 9Servicio de Hematología, Hospital La Fe, 46026 Valencia, Spain; randreu69@gmail.com; 10Servicio de Medicina Nuclear, Hospital Virgen de la Arrixaca, 30120 Murcia, Spain; laura.frutos@yahoo.es (L.F.); jlnf@um.es (J.L.N.); 11Servicio de Medicina Nuclear, Hospital Virgen del Rocío, 41013 Sevilla, Spain; rosalvarezp@hotmail.com; 12Servicio de Medicina Nuclear, Hospital 12 de Octubre, 28041 Madrid, Spain; psft24@gmail.com; 13Servicio de Medicina Nuclear, Hospital Universitari de Bellvitge-IDIBELL, 08908 Barcelona, Spain; montserrat.cortes.idi@gencat.cat; 14Servicio de Medicina Nuclear, Hospital Clínico Universitario de Salamanca/IBSAL, 37007 Salamanca, Spain; ptamayo@usal.es; 15Servicio de Medicina Nuclear, Hospital Universitario de Canarias, 38320 La Laguna, Spain; junagor@gobiernodecanarias.org; 16Servicio de Medicina Nuclear, Hospital Ramón y Cajal, 28034 Madrid, Spain; amlorca@salud.madrid.org; 17Servicio de Medicina Nuclear, Hospital La Fe, 46026 Valencia, Spain; cristinaruiz.mac@gmail.com

**Keywords:** mantle cell lymphoma, PET/CT, bone marrow biopsy

## Abstract

This research explores the role of different methods for assessing bone marrow infiltration (BMI) in mantle cell lymphoma (MCL) during initial diagnosis. Traditional bone marrow biopsy (BMB) is a standard procedure, but newer imaging techniques like PET/CT scans may offer additional or alternative insights. Our study aims to clarify the diagnostic accuracy and prognostic value of these methods, both individually and in combination, for predicting patient outcomes. We propose a new prognostic model that integrates PET/CT results, which could improve the ability to classify patients by risk. The findings may impact clinical approaches by guiding better-informed decisions in MCL diagnosis and prognosis.

## 1. Introduction

In the initial evaluation of mantle cell lymphoma (MCL), analysis of bone marrow (BM) infiltration (BMI) presents a complex scenario in which BMI assessed by BM biopsy (BMB) is not significant by either the widely used MIPI [1], or the more refined updated versions MIPI-c and MIPI-g [2,3]. Despite this clear statement, guidelines currently provide inconsistent recommendations regarding BM evaluation by biopsy/aspiration in upfront staging. Interestingly, a similar situation exists for positron emission tomography/computed tomography (PET/CT). In this regard, the NCCN guidelines (V3 2024) [4] consider PET/CT essential and BMB useful only in limited circumstances. Conversely, current British guidelines recommend BMB, but not PET/CT due to its lack of sensitivity [5]. On the other hand, the current Spanish GELTAMO guidelines [6] consider both procedures mandatory.

The above discrepancies reflect the results of MIPI/MIPIc-g on the one hand, and the different data reported for PET or PET/CT BMI analysis [7,8], which are ultimately based on the heterogeneous and complex biology of MCL [9,10]. For PET/CT, this translates into uneven 18F-fluorodeoxyglucose (FDG) avidity and therefore low sensitivity for detecting both BM and extranodal lesions, as well as variable inter- and intra-individual FDG avidity [7,8,11,12,13,14].

Even more imprecise is the use of both cytomorphologic and flow cytometric (FC) analysis of BM aspirates (BA) in this setting. According to the NCCN guidelines [4], FC (from either BM or peripheral blood) is considered essential, while BA is considered optional in the BM evaluation. Similarly, current British guidelines recommend which markers should be analyzed by either immunohistochemistry or FC, but do not specify whether they are mandatory or recommended [5,15].

All of the above issues are critical because treatment decisions can be based either on stage (i.e., NCCN guidelines [4]), where BMI may be relevant or on risk management, where it is not with current indices [16].

In the frontline evaluation of B-cell non-Hodgkin’s lymphoma (NHL), and based on our own previously published experience [11,17,18], and the current GELTAMO guidelines [6], either PET/CT and BMB are performed, the latter always simultaneously with a BA. In addition to the morphologic study of BA, samples are obtained for additional immunophenotypic, cytogenetic (karyotype and/or FISH) and molecular analysis in cases where lymphoma infiltration is observed or strongly suspected; this approach aims to integrate this data set according to WHO recommendations [19].

Based on the above, we herein consider a distinct and more holistic approach for the initial assessment of BMI in MCL-NHL. To this end, we have evaluated PET/CT on the one hand and BMB, BA and FC on the other hand. With the aim of clarifying the exact role of each of these modalities, we recruited a meaningful cohort of patients and focused on analyzing both the accuracy and the clinical impact of each modality or combination of modalities. In addition, our results have led us to propose an innovative prognostic index for progression-free survival (PFS) that incorporates BMI using PET/CT.

## 2. Patients and Methods

### 2.1. Our Study Is a Retrospective Analysis with Consecutive Recruitment of Patients with the Following Characteristics

#### 2.1.1. Patients

Patients > 18 years of age with a diagnosis of MCL according to the WHO classification between 2007 and 2022, with both BMB and PET/CT performed at baseline, from eight tertiary centers in Spain were included. Patients had not received chemotherapy or corticosteroids. The pathology and PET/CT results were blinded to the professionals interpreting either set of results.

This study was approved by the Morales Meseguer University Hospital IRB (EST:08/14) and was conducted in accordance with the Declaration of Helsinki.

#### 2.1.2. Bone Marrow Biopsy

In Spain, unguided unilateral posterior iliac crest BMB and BA are recommended in patients diagnosed with NHL according to the GELTAMO guidelines, although there is no consensus for MCL. According to national pathology guidelines, CD20, CD5, CD3 and CCND1 were used to confirm B-cell infiltration and to exclude reactive mixed nodules.

BMB, BA and FC were evaluated by experienced hematopathologists at each center. For FC, at least CD19, CD20, CD5, CD23, FMC7, and surface light chain were evaluated [20]. Results were obtained from individual reports and were not subsequently verified. Data from molecular tests were not used in the present work.

#### 2.1.3. PET/CT Imaging and Analysis

PET/CT studies were performed with the following PET/CT scanners: Gemini TF64, Gemini GXL, and Gemini TF16 (Philips Gemini, Andover, MA, USA), Discovery LS, Discovery ST, Discovery STE, and Discovery IQ (GE Healthcare Discovery, Waukesha, WI, USA), and either Biograph mCT 20 Flow, Biograph TP16, and Biograph 6 (Siemens Biograph, Erlangen, Germany). Procedures, quality control and interpretation guidelines are detailed in our previous works [11,17].

Visual analysis of BMI by PET/CT was considered positive in the presence of unifocal (single lesion), bifocal, multifocal (≥3 lesions) or focal lesions with diffuse uptake exceeding that of the liver, which could not be explained by benign findings on the underlying CT scan or patient’s medical history (i.e., fractures). Purely diffuse FDG uptake was not considered related to BMI, as previously reported by other authors [12], due to anemia being the main potential paraphysiological cause of increased BM uptake in these patients. On the other hand, for the survival analysis, the SUVmax, whether nodal or extranodal, was considered, with a cut-off of 9 according to the ROC analysis on the total cohort. A detailed description of PET/CT analysis methods was previously stated elsewhere [11].

#### 2.1.4. Statistics

We used the Kaplan–Meier and Cox methods to analyze overall survival (OS) and PFS, with a two-sided *p*-value < 0.05 for a factor in the univariate analysis to be included in the multivariate analysis, where a *p*-value < 0.05 was considered statistically significant. To avoid collinearity in the multivariate analysis, we decomposed the biological IPI-NCCN, MIPI, and MIPI-c into their underlying factors. We considered BMI through different measures in four models: BMB, PET/CT, BMB & PET/CT, and BMBorPET/CT. In Kaplan–Meier, log-rank survival plots and in Cox methods, partial residuals were examined to assess whether the underlying assumption of proportional hazards was met. Two cohorts were analyzed: the total cohort (*n* = 148) and intensively treated MCL patients (*n* = 128). In the intensively treated group, we included patients treated with immunochemotherapy: R-CHOP (Rituximab, Cyclophosphamide, Doxorubicin, Vincristine, and Prednisone), R-CHOP-DHAP (R-CHOP alternating with Rituximab, Dexamethasone, Cytarabine, and Cisplatin), including those in the R-CHOP-DHAP regimen who also received Ibrutinib in the TRIANGLE clinical trial, RB (Rituximab and Bendamustine), and R-HYPER-CVAD (Cyclophosphamide, Vincristine, Adriamycin, Methotrexate, Cytarabine, and Dexamethasone).

The accuracy of the assays was evaluated as previously described [11]. The final diagnosis of BMI by lymphoma was made in cases of positive PET/CT (PET/CT+) or positive BMB (BMB+). Statistical analysis was performed using SPSS software (IBM SPSS Statistics 21, IBM Corporation, Chicago, IL, USA), R-4.4.1 (https://www.r-project.org/, accessed on 20 January 2024) and Epidat (http://dxsp.sergas.es, accessed on 20 January 2024).

## 3. Results

### 3.1. Patient Characteristics

The results of 148 patients were analyzed. The main characteristics at baseline are shown in Table 1. With a median age at diagnosis of 63 years (interquartile range, 53–70 years), the majority of patients (87%) received upfront immunochemotherapy.

### 3.2. Performance of PET/CT and BMB Findings in Staging

PET/CT was positive for BMI in 33 patients and negative in 115 patients. Among the positive patients, 27 also had BMB+, while 83 patients had BMB+ with a negative PET/CT. BMB was positive in 110 patients and negative in 38. Of the negative patients, 6 had PET/CT+ (Table 2).

Focusing on the performance of PET/CT, the sensitivity was 28.45 (95% confidence interval (CI); 19.81–37.09), the negative predictive value (NPV) was 27.83 (95% CI; 19.20–36.45), and the accuracy was 43.92 (95% CI; 35.59–52.25). For BMB, the sensitivity was 94.83 (95% CI; 90.37–99.29), the NPV was 84.21 (95% CI; 71.30–97.12), and the accuracy was 95.95 (95% CI; 92.43–99.46) (Table 3). Considering BMB as reference for BMI, the use of PET/CT upstaged 32 patients (21.6%) to Ann Arbor IV. However, when excluding cases with extramedullary involvement on PET-CT, BMI by PET-CT would have upstaged only 17 patients (11.5%), all of whom also had BMI infiltration by BMB. On the other hand, without extramedullary disease or BMI by PET-CT, BMB would have upstaged 61 patients (41.2%) to Ann Arbor IV.

### 3.3. Impact of PET-CT and BMB Findings on Survival

#### 3.3.1. Whole Cohort: Deconstructed Prognostic Scores

With a median follow-up of 51 months (range, 1.3–185.8), 40 patients (27%) experienced disease progression and 34 (23%) died, 16 of them due to lymphoma. Univariate analysis of OS and PFS is shown in Table 4A.

LDH above the upper normal limit (UNL), hemoglobin less than 120 g/L, male sex, age older than 70 years, and SUVmax ≥ 9 were significantly associated with shorter PFS in univariate analysis. Among the different definitions of BMI, PET-CT+ and combined PET/CT&BMB+ were significantly associated with shorter PFS in univariate analysis. Two multivariate models were constructed (one for each of the significant BMI measures) (Table 5A). A PET-CT+ and the combined PET/CT&BMB+ added independent prognostic value for PFS.

Regarding OS, hemoglobin less than 120 g/L and age older than 70 years were significantly associated with shorter OS in univariate analysis, whereas Ki-67 > 30% in bone marrow, lymph nodes or other tissues showed a trend toward statistical significance. Among the different definitions of BMI, a BMB+, a PET/CT+, the combined PET/CTorBMB+, and the combined PET/CT&BMB+ were significantly associated with shorter OS in univariate analysis. Four multivariate models were created (one for each of the significant BMI measures) (Table 5A); of the above four, only BMI infiltration by PET-CT remained significant for shorter OS.

#### 3.3.2. Intensively Treated MCL Cohort: Deconstructed Prognostic Scores

With a median follow-up of 52.4 months (range, 1.3–185.8), 35 patients (27.3%) progressed and 28 (21.9%) died, 13 of them due to lymphoma. Univariate analysis of OS and PFS is shown in Table 4B.

LDH above ULN, male gender and age older than 70 years were significantly associated with shorter PFS. In addition, PET-CT+ and the combined PET/CT&BMB+ were also significantly associated with shorter PFS in univariate analysis. Two multivariate models were constructed (Table 5B). PET-CT+ and the combined PET/CT&BMB+ added independent prognostic value.

Regarding OS, hemoglobin less than 120 g/L and age older than 70 years were significantly associated with shorter OS in univariate analysis. Among the different definitions of BMI, BMB+, PET/CT+, the combined PET/CTorBMB+ and the combined PET/CT&BMB+ were significantly associated with shorter OS in univariate regression. Four multivariate models were constructed (Table 5B). From the above, PET-CT+ and the combined PET/CT&BMB+ remained significant for shorter OS.

### 3.4. PFS Prognostic Model Score Calculation-Mantle Cell Lymphoma PET/CT Index (MCL-PET-I)

We attempted to translate our findings into a new coherent prognostic model that included three of the most significant parameters from our multivariate model using the Cox method on PFS: LDH above UNL, age older than 70 years, and BMI by PET-CT. For this purpose, we assigned a score of 1 to each of these parameters after weighting the hazard ratio. Patients with composite scores of 0 (*n* = 57, 38.5%), 1 *(n* = 69, 46.6%), and ≥2 (*n* = 22, 14.9%) were assigned to low-risk, intermediate-risk, and high-risk groups, respectively.

The 5-year PFS rates were 90%, 60% and 25% for the low-risk, intermediate-risk and high-risk cohorts, respectively (*p* < 0.0001) (Figure 1A). When the same parameters were analyzed for 5-year OS, the rates were 90%, 70% and 50% for the low-risk, intermediate-risk and high-risk cohorts, respectively (*p* = 0.00025) (Figure 1B).

## 4. Discussion

For decades, BMI assessment by BMB has been central to the upfront evaluation of lymphoma. This was because BMI led to changes in staging and thus prognosis, assuming the Ann Arbor mindset in which bone marrow histology was the “gold standard” and indeed the only technique for this purpose.

Although MCL is a relatively “new” entity, the above statement was assumed for years in its diagnostic workup, as they were likely different prognostic indices, despite their poor clinical translation [21,22,23,24,25,26,27,28]. It was only with the introduction of MIPI that BMI via BMB was discarded as a prognostic parameter due to its poor significance [1]. A further step in this puzzle is the introduction of PET/CT. Regarding this technique, some published reports showed that upfront PET/CT could contribute to prognosis [8], while others did not [12]. The above is reflected in the inconsistent recommendations shown in different guidelines: while some of them maintain BMB as a mandatory test, others do not, with a likely situation occurring with PET/CT. Therefore, the questions at this point are: What is the basis for this situation? Is BMI really worth analyzing in MCL? And, if so, with which currently used technique?

In the present paper, we have approached the initial assessment of BMI in a large series of MCL from a novel point of view. In this regard, we have considered the value of BMB, BA and FC on the one hand and PET/CT on the other hand.

In our series, 116 patients out of 148 (78% of the whole series) showed BMI; of these 116, 110 (95%) were BMB+, while additionally only 6 (5%) were PET/CT+ and BMB negative. PET/CT was positive in 27 patients who were BMB+. This translates to NPV, sensitivity and accuracy of PET-CT and BMB of 27.8% vs. 84.2%, 28.4% vs. 94.8% and 43.9% vs. 95.9%, respectively. These results are close to those reported in previous series [29] and indicate that PET/CT is much less sensitive and accurate than BMB and therefore insufficient to replace this technique for BMI analysis with staging purposes, as clearly indicated by the fact that 41.2% of patients are stage IV BMB+ and PET-CT negative. Similarly, the low sensitivity of PET/CT for BMI has also been reported by our group and others in follicular lymphoma, reinforcing that PET/CT alone may not be sufficient for BMI detection in lymphomas with low metabolic activity [18,30]. This stands in contrast to Hodgkin’s lymphoma and diffuse large B-cell lymphoma, where PET/CT may substitute BMB in certain particular settings according to current guidelines, further emphasizing the heterogeneity across lymphoma subtypes [31]). From another perspective, others had suggested avoiding BMB in those MCL patients with PET/CT+ [12], in fact, a minority of patients. From our point of view, this could only be considered in those selected patients within this subgroup in whom the entire cytogenetic and molecular workout had already been performed in a distinct sample (i.e., lymph node suspension). In the remaining cases, i.e., the majority, where BMI by PET/CT is negative, BMB remains indispensable for accurate staging, especially in patients with high clinical suspicion, as negative results do not reliably exclude marrow infiltration.

How did the above findings translate into clinical consequences? Interestingly, and for the univariate analysis considering the whole series, age, hemoglobin, PET/CT and PET/CT&BMB were significant for both PFS and OS, while in the intensively treated group, age, PET/CT and PET/CT&BMB were significant for both. In addition, other parameters, such as LDH (with a strikingly high significance for PFS), gender, BMB and PET/CTorBMB, showed clinical value in both series and were therefore included in the multivariate analysis. In this analysis, only age and PET/CT were significant for PFS and OS in both series, with additional value for LDH (again with a striking significance), sex, hemoglobin, and PET/CT&BMB for PFS. Of note, while age and hemoglobin were relevant as expected based on MIPI statements [1,32], WBC, another MIPI parameter, was not significant in either group, nor were B2-microglobulin, a parameter from FLIPI2, Ki-67 and blastoid morphology. We have no clear explanation for our negative findings regarding WBC, although conflicting data on the value of MIPI in a variety of clinical settings have been reported previously [33,34,35]. Regarding Ki-67 and blastoid morphology, we must point out that our results may be confounded by the fact that only half of the series could be analyzed. Notable is the lack of prognostic value of BA and FC: their added value to BMB for prognosis is marginal, as it is for staging, although their interest could be based both in the possible convenience of being easily obtained samples for genetic/molecular characterization of the tumor and sharpening the detection of blastoid morphology. For now, as suggested by Scheubeck et al. [3], it seems prudent to maintain BA analysis.

Of particular interest are our findings regarding the clinical impact of BMI evaluation by PET/CT and BMB. Our data indicate that visual analysis of PET/CT for BMI shows a strong significance for both PFS and OS, in agreement with previous reports that considered visual analysis with or without IHP guidelines or Deauville criteria [12,36]. This is in contrast to the marginal value of BMB, which only shows significance when combined with PET/CT. How can this qualitatively different effect be explained if BMB is a much more sensitive technique for the detection of BMI according to our own results and those previously reported by others? In our opinion, two considerations must be taken into account. First, as mentioned above, BMB is very sensitive—the majority of patients show BMI when using it; this could probably be the problem: BMB does not seem to be a good technique to discriminate the behavior of MCL. In contrast, PET/CT is not as sensitive, although it seems to be quite related to tumor cell biology, probably as a consequence of a distinctive 18FDG uptake. Our results show that when used for BMI, PET/CT detected those cases that did worse in both PFS and OS, thus acting as a surrogate marker of aggressiveness. Interestingly, this was not the case when considering extranodal/extramedullary stage IV by PET/CT. Again, we do not have a clear explanation for the latter, although it may ultimately reflect a heterogeneous and distinct tumor biology in both locations, as mentioned previously [7,8,9,10].

To date, the impact of BMI on prognostication with PET/CT has shown inconsistent results [7]. In this regard, and contrary to what was suggested by the authors [12], in our series, the value of BM PET/CT for prognosis by upstaging was marginal. On the contrary, our data point to a distinct interpretation of baseline BMI by PET/CT, which could be more related to those authors suggesting an intrinsic value of the technique, either considered alone or in combination with MIPI; in this regard, it is worth mentioning the interesting, but not widely applied, proposal of voxel analysis by Morgan et al. [37]. However, and in our opinion more clinically relevant, are the data from the LyMa-PET Project [8]. Although MIPI was the first prognostic index specific for MCL patients, its validation has been inconsistent [38]. Owing to this, a very interesting attempt has been made to combine the highest PET/CT SUVmax with MIPI, even establishing an SUVmax cutoff of 10.3, importantly, in any anatomical location [8]. This suggestion was based on the hypothesis that the prognosis of MCL is based on the most aggressive tumor area within the whole economy [39,40]. In this regard, and interestingly, in our series, SUVmax turned out to be significant only for PFS in the whole series. Other proposals for MIPI refinement have been proposed, including its combination with Ki-67 and pleomorphic or blastoid morphology, although again with varying agreement [2,27]. More recently, the introduction of the analysis of *TP53* by both genetic and molecular techniques has further improved MIPI. In the end, all the above proposals aim to complement the “classical” MIPI data with different surrogate markers of aggressiveness and are mostly limited by inter-individual tumor heterogeneity, sampling and analysis standardization and, although of great interest, only Ki-67 and *TP53* mutations have been validated so far [41]. Differently and as a consequence of our results, especially on WBC and BMI-PET/CT, and the pronounced clinical impact of either age or LDH, we elaborated a proposal for the new MCL-PET-I that avoids MIPI. The composite scores using these three readily available parameters allow differentiation between three groups with consistently different clinical outcomes, and we extrapolate this to OS, again with good differentiation between the three groups.

At present, the identification of high-risk patients, whether BMI-PET/CT+ or based on previously described prognostic markers such as *TP53* mutations or high Ki-67 expression, does not translate into a change in therapeutic strategy. Recently, the TRIANGLE trial demonstrated that adding ibrutinib to first-line treatment resulted in superior efficacy in younger MCL patients. However, this was accompanied by increased toxicity, particularly when administered after ASCT [42]. Unfortunately, the number of patients treated with this regimen in our study is too low to draw meaningful conclusions.

Nonetheless, the identification of high-risk patients using PET/CT, particularly those classified in the high-risk group according to the MCL-PET-I index, underscores the need for prospective, controlled studies to determine whether intensified or novel-agent-based first-line therapies could improve outcomes. Comparative studies should consider current standard regimens as controls and assess the addition of agents like Bruton’s tyrosine kinase inhibitors or other targeted therapies to improve the prognosis of these high-risk subsets.

We acknowledge that our study design, retrospective and multicenter, has both strengths and weaknesses and may raise a number of issues. First, although a multicenter approach may be closer to “real life” clinical practice, its main limitation in the particular MCL setting is the lack of standard validated criteria for both BMB and PET/CT evaluation [12]. Indeed, this possible center heterogeneity has led to inconsistent results in previous studies and could, therefore, be considered as a confounding factor [43,44,45]. Second, patients considered “intensively treated” were not uniformly medicated. Third, we did not differentiate between age categories due to sample size. Fourth, Ki-67, blastoid morphology, *TP53* and SUVmax could not be properly analyzed due to missing data; in our opinion, all four merit further consideration; in this regard, the study of their combined value with visual BMI-PET/CT is warranted. Furthermore, and interestingly, WBC, a validated MIPI parameter, was not found to be significant in our series.

The recent emergence of genetically significant data has further unveiled the complexity of MCL prognosis [3,16,41,46,47]. We believe that in the near future, routine detailed genetic studies will be required to properly guide targeted therapies [16]. This, in turn, is likely to lead to the addition of genetic risk to clinical indices and thus to a reassessment of prognosis. In the meantime, it may be time to incorporate PET/CT, a widely used technique today, into MCL prognostic indices. As mentioned, previous studies have suggested the value of PET/CT, either using visual analysis or SUVmax as a complement to MIPI/MIPI-c [8]. Distinctly, our proposal circumvents those indices focused in BMI-PET/CT based on the hypothesis of a specific and distinct value of its analysis as a surrogate marker of aggressive behavior, as has been previously suggested [12]. The fact of bypassing these corroborated indices makes mandatory their validation through the analysis of a larger series, which is our next objective.

## 5. Conclusions

Our data indicate that in the upfront work of MCL, BMB is necessary for staging, while BMI-PET/CT confers a marked prognostic significance. In our opinion, both techniques are currently necessary in this setting. In addition, the BMI-PET/CT results in our series lead us to propose the MCL-PET-I prognostic index, which allows reliable differentiation between clinical groups.

## Figures and Tables

**Figure 1 cancers-16-04189-f001:**
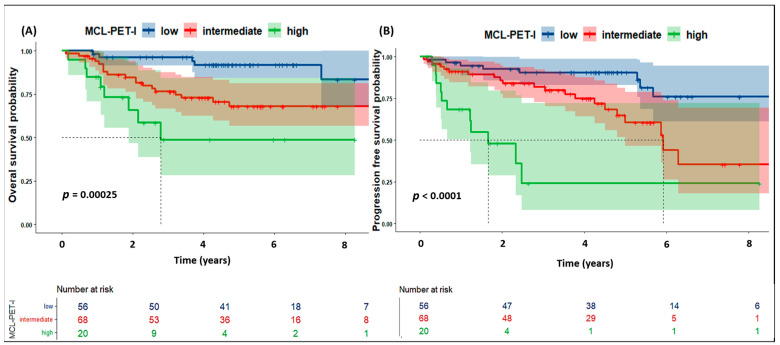
Overall (**A**) and progression-free survival (**B**) according to the MCL-PET-I prognostic model. Risk categories: Low (Score = 0), Intermediate (Score = 1), and High (Score ≥ 2). The prognostic score is based on LDH > UNL, Age > 70 Years, and PET/CT bone marrow involvement (1 Point Each).

**Table 1 cancers-16-04189-t001:** Patient characteristics.

Characteristic	*N* = 148, n (%)
Age > 70 y	39 (26.3)
Male sex	116 (78.4)
Subtype	
Conventional MCL	132 (89.2)
Leukemic non-nodal MCL	16 (10.8)
ECOG ≥ 2	6 (4)
Ann Arbor Stage	
I–II	10 (6.7)
III	10 (6.8)
IV	128 (86.5)
Nodal sites involvement > 4	90 (60.1)
Extranodal sites involvement (other than bone marrow)	9 (6)
LoDLIN > 6 cm	22 (14.8)
Albumin < 40 g/L	41 (27.7)
LDH > UNL	92 (62.1)
β2m > UNL	69 (46.6)
Hemoglobin < 120 g/L	40 (27)
WBC > UNL	44 (29.7)
SUVmax ≥ 9	34 (22.9)
Blastoid morphology	8 (5.4)
Ki-67 ≥ 30%	49 (33.1)
*TP53* mutated	7 (4.7)
MIPI score	
Low	99 (66.8)
Intermediate	33 (22.2)
High	16 (10.8)
MIPI-c score	
Low	41 (27.7)
Low-intermediate	62 (41.8)
High-intermediate	10 (6.7)
High	3 (2)
**Treatment and outcome**	
Induction treatment	
R-CHOP/R-DHAP	53 (35.8)
R-CHOP/R-DHAP + ibrutinib (TRIANGLE trial)	18 (12.1)
R-CHOP	27 (18.2)
R-bendamustine	14 (9.4)
R-HyperCVAD	16 (10.8)
R-ibrutinib	8 (5.4)
Observation	1 (0.6)
Others *	11 (7.4)
ASCT	59 (39.8)
Allo-HSCT	8 (5.4)
Rituximab maintenance	83 (56)
Response	
Complete response	119 (80.4)
Partial response	16 (10.8)
Stable disease	2 (1.3)
Progressive disease	5 (3.3)
Not evaluable	6 (4)
5-y PFS, % (95% CI)	58.2 (48.7–66.5)
5-y OS, % (95% CI)	75.1 (66.5–81.9)
Follow-up in months, median (range)	51 (1.3–185.8)

Missing data: blastoid morphology (*n* = 47), Ki-67 (*n* = 32), MIPI-c (*n* = 32), SUVmax (*n* = 11), *TP53* mutation status (*n* = 90). * R-cyclophosphamide (*n* = 5), VR-CAP (*n* = 5), R-BAC (*n* = 1). Allo-HSCT, allogenic hematopoietic stem cell transplantation; ASCT, autologous stem cell transplantation; β2m, Beta2 microglobulin protein; CI, cumulative incidence; ECOG, Eastern Cooperative Oncology Group; LDH, lactate dehydrogenase; LoDLIN, longest diameter of the largest involved node; MCL, mantle cell lymphoma; OS, overall survival; PFS, progression-free survival; UNL, upper normal limit; WBC, white blood cells.

**Table 2 cancers-16-04189-t002:** Diagnostic Performance of PET/CT and BMB for Detecting Bone Marrow Involvement.

	BMB Negative	BMB Positive	Total
PET/CT negative	32	83	115
PET/CT positive	6	27	33
Total	38	110	148

BMB, Bone marrow biopsy.

**Table 3 cancers-16-04189-t003:** Diagnostic performance of PET/CT and bone marrow biopsy in terms of sensitivity, negative predictive value and accuracy for detecting bone marrow involvement.

	BMI Evaluation Method	
	PET/CT	BMB
Sensitivity (%)	28.45 (19.81–37.09)	94.83 (90.37–99.29)
Negative predictive value (%)	27.83 (19.20–36.45)	84.21 (71.30–97.12)
Accuracy (%)	43.92 (35.59–52.25)	95.95 (92.43–99.46)

BMB, Bone marrow biopsy; BMI, bone marrow involvement.

**Table 4 cancers-16-04189-t004:** Univariate analysis of factors influencing survival outcomes—Logrank test.

A. Whole Cohort, *n* = 148					
**Variable**	***N* (%)**	**PFS**		**OS**	
		**HR (95% CI)**	** *p-* ** **Value**	**HR (95% CI)**	** *p-* ** **Value**
Age > 70 y	39 (26.3)	2.14 (1.1–4.16)	**0.025**	2.37 (1.18–4.78)	**0.015**
Male sex	116 (78.4)	3.84 (1.18–12.5)	**0.025**	1.06 (0.46–2.45	0.888
LDH > UNL	92 (62.1)	2.87 (1.52–5.41)	**0.001**	1.59 (0.78–3.24)	0.197
WBC > UNL	44 (29.7)	0.86 (0.42–1.72)	0.671	1.08 (0.51–2.28)	0.833
Ki-67 > 30%	49 (33.1)	1.09 (0.58–2.04)	0.786	1.83 (0.92–3.63)	0.082
β2m > UNL	69 (46.6)	1.31 (0.71–2.45)	0.387	1.44 (0.73–2.86)	0.239
Hemoglobin < 120 g/L	40 (27)	1.96 (1.01–3.81)	**0.049**	2.71 (1.35–5.34)	**0.005**
SUVmax ≥ 9	34 (22.9)	1.73 (0.91–3.32)	**0.011**	2.15(1.11–4.54)	0.231
Blastoid morphology	8 (5.4)	2.45 (0.86–7.04)	0.09	1.22 (0.28–5.19)	0.782
PET/CT BMI+	33 (22.2)	3.11 (1.55–6.24)	**0.001**	2.82 (1.35–5.87)	**0.006**
BMB positive	110 (74.3)	1.83 (0.76–4.01)	0.17	3.72 (1.13–12.19)	**0.030**
BA/FC positive	103 (69.5)	0.99 (0.53–1.84)	0.97	1.67 (0.81–3.34)	0.167
PET/CT stage IV (extramedullary disease)	49 (33.1)	1.28 (0.66–2.46)	0.45	0.93 (0.44–1.05)	0.853
PET/CT&BMB positive	27 (18.2)	3.41 (1.65–7.03)	**0.001**	2.95 (1.39–6.26)	**0.005**
PET/CTorBMB positive	116 (78.3)	1.78 (0.75–4.26)	0.19	4.91 (1.17–20.53)	**0.029**
B. Intensively treated cohort, *n* = 128					
**Variable**	***N* (%)**	**PFS**		**OS**	
		**HR (95% CI)**	** *p-* ** **Value**	**HR (95% CI)**	** *p-* ** **Value**
Age > 70 y	23 (17.9)	2.14 (1.04–4.38)	**0.037**	2.48 (1.10–5.57)	**0.028**
Male sex	106 (82.8)	4.24 (1.01–17.61)	**0.048**	1.31 (0.45–3.79)	0.621
LDH > UNL	42 (32.8)	2.86 (1.46–5.61)	**0.002**	1.68 (0.78–3.63)	0.184
WBC > UNL	30 (23.4)	1.03 (0.48–2.19)	0.947	0.79 (0.34–1.80)	0.570
Ki-67 > 30%	47 (36.7)	1.00 (0.50–1.96)	0.985	1.90 (0.89–4.07)	0.101
β2m > UNL	55 (42.9)	1.10 (0.56–2.14)	0.784	1.13 (0.53–2.42)	0.755
Hemoglobin < 120 g/L	35 (27.3)	1.90 (0.94–3.83)	0.073	2.34 (1.08–5.05)	**0.031**
SUVmax ≥ 9	30 (23.4)	1.90 (0.86–4.20)	0.113	1.76 (0.72–4.29)	0.211
Blastoid morphology	7 (5.4)	1.72 (0.52–5.73)	0.371	0.77 (0.10–5.78)	0.791
PET/CT BMI positive	31 (24.2)	2.87 (1.36–6.08)	**0.006**	3.02 (1.37–6.66)	**0.006**
BMB positive	93 (72.6)	1.88 (0.78–4.52)	0.161	3.15 (0.95–10.47)	0.061
BA/FC positive	89 (69.5)	1.08 (0.55–2.09)	0.832	1.80 (0.82–3.92)	0.141
PET/CT stage IV (extramedullary disease)	31 (24.2)	1.20 (0.60–2.43)	0.606	0.89 (0.39–2.03)	0.782
PET/CT&BMB positive	25 (19.5)	3.12 (1.45–6.70)	**0.004**	3.13 (1.39–7.03)	**0.006**
PET/CTorBMB positive	100 (78.1)	1.95 (0.76–5.02)	0.168	4.12 (0.97–17.39)	**0.050**

Significant *p*-values are in bold type. Missing data (A) in the whole cohort: blastoid morphology (*n* = 47), Ki-67 (*n* = 32), SUVmax (*n* = 11); (B) in the intensively treated cohort blastoid (*n* = 44), Ki-67 (*n* = 26), SUVmax (*n* = 8). β2m, Beta2 microglobulin protein; BMB, bone marrow biopsy; BMI, bone marrow involvement; BA/FC, bone marrow aspiration/flow cytometry; CI, confidence interval; HR, hazard ratio; LDH, lactate dehydrogenase; OS, overall survival; PFS, progression-free survival; UNL, upper normal limit; WBC, white blood cells.

**Table 5 cancers-16-04189-t005:** Multivariate analysis of factors influencing survival outcomes according to different BMI evaluation methods—Cox proportional hazards model.

A. Whole Cohort, *n* = 148												
**BMI Evaluation Method**	**PET/CT**				**BMB**		**PET/CT&BMB**				**PET/CTorBMB**	
**Variable**	**PFS**		**OS**		**OS**		**PFS**		**OS**		**OS**	
	**HR**	** *p* **	**HR**	** *p* **	**HR**	** *p* **	**HR**	** *p* **	**HR**	** *p* **	**HR**	** *p* **
LDH > UNL	2.99 (1.47–6.06)	**0.002**					2.94 (1.45–5.96)	**0.003**				
Hb < 120 g/L	1.09 (0.47–2.5)	0.833	1.96 (0.85–4.48)	0.11	2.22 (1.08–4.54)	**0.030**	1.06 (0.45–2.46)	0.899	0.1.98 (0.85–4.62)	0.110	2.38 (1.17–4.81)	**0.016**
Male sex	4.27 (1.27–14.28)	**0.018**					4.29 (1.28–14.28)	**0.018**				
Age > 70 y	2.89 (1.48–5.65)	**0.002**	2.67 (1.31–5.45)	**0.007**	2.54 (1.25–5.14)	**0.009**	2.76 (1.42–5.39)	**0.003**	2.58 (1.27–5.23)	**0.008**	2.55 (1.26–5.16)	**0.009**
BMI	2.66 (1.11–6.46)	**0.027**	2.28 (1.31–5.45)	**0.042**	2.71 (0.81–9.08)	0.108	2.85 (1.14–7.11)	**0.025**	2.17 (0.86–5.45)	0.099	4.15 (0.98–17.61)	0.053
B. Intensively treated cohort, *n* = 128												
**BMI Evaluation Method**	**PET/CT**				**BMB**		**PET/CT&BMB**				**PET/CTorBMB**	
**Variable**	**PFS**		**OS**		**OS**		**PFS**		**OS**		**OS**	
	**HR**	** *p* **	**HR**	** *p* **	**HR**	** *p* **	**HR**	** *p* **	**HR**	** *p* **	**HR**	** *p* **
LDH > UNL	3.11 (1.50–6.45)	**0.002**					3.06 (1.48–6.34)	**0.003**				
Hb < 120 g/L	1.16 (0.51–2.68)	0.721	1.48 (0.61–3.60)	0.383	1.99 (0.91–4.35)	0.086	1.13 (0.48–2.64)	0.769	1.31 (0.56–3.26)	0.562	1.95 (0.89–4.26)	0.094
Male sex	5.26 (1.21–22.73)	**0.030**					5.26 (1.21–22.51)	**0.026**				
Age > 70 y	3.22 (1.52–6.82)	**0.002**	2.91 (1.27–6.70)	**0.021**	2.40 (1.07–5.40)	**0.034**	3.08 (1.46–6.49)	**0.003**	2.77 (1.21–6.31)	**0.016**	2.46 (1.09–5.53)	**0.030**
BMI	2.68 (1.11–6.53)	**0.030**	2.96 (1.16–7.52)	**0.022**	2.53 (0.74–8.55)	0.135	2.84 (1.13–7.13)	**0.026**	2.87 (1.08–7.58)	**0.033**	3.56 (0.83–13.26)	0.087

Significant *p*-values are in bold type. BMB, bone marrow biopsy; BMI, bone marrow involvement; Hb, hemoglobin; HR, hazard ratio (with 95% confidence interval); LDH, lactate dehydrogenase; OS, overall survival; PFS, progression-free survival; UNL, upper normal limit.

## Data Availability

All data generated during this study are included in this published article. The raw data analyzed during the current study are available from the corresponding author on reasonable request.

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
