# Peer review of "The Value of Bone Marrow Assessment by FDG PET/CT, Biopsy and Aspirate in the Upfront Evaluation of Mantle Cell Lymphoma: A Nationwide Cohort Study"

_cancers, 2024, doi:10.3390/cancers16244189_

Round 1
Reviewer 1 Report
Comments and Suggestions for Authors
Ródenas-Quiñonero et al. report on a retrospective study of pretherapeutic FDG-PET/CT in 148 MCL patients investigated over a 15 year period at 8 centers in Spain. This is a considerably large patient cohort and language is of good quality.
The first and foremost result demonstrated in this study is that PET/CT is not sufficient to detect bone marrow infiltration (BMI) from MCL (NPV 28%). This is in keeping with previous findings. This finding is even more critical as the incidence of BMI was very high in the current cohort: When PET and bone marrow biopsy (BMB) are combined as the reference standard, the incidence of BMI was as high as 78% (116/148 cases; only 6 of which were detected exclusively by PET).
As based on inferior PFS and OS in the subgroup of PET+ MCL, the authors propose to include PET/CT into a prognostic model.
Specific comments:
This reviewer suggests to add information on patient selection for PET/CT. Was it performed in all patients with MCL? Add like a flow chart diagram to RESULTS to visualize recruitment, drop outs, missing data etc.
Please add to RESULTS: What where the criteria for PET positivity? There is merely a statement saying “visual analysis”. Did the investigators include any semiquantitative criteria like SUVs? How about reference organs like Liver, Bloodpool and normal Bone marrow uptake?
This reviewer suggests to strengthen the DISCUSSION as follows:
As long as knowledge of BMI is considered necessary for staging in MCL, PET/CT cannot substitute BMB.
It should be pointed out, that based on the results reported, the clinical gain of PET/CT for MCL patients is not clear. Obviously, PET/CT identifies a subgroup of MCL patients with particularly aggressive and/or high volume BM disease. But could this be turned into a benefit for patients that justifies to perform this relatively expensive imaging test in all MCL patients? Specifically, could the authors propose a prospective, controlled study that evaluates intensified first-line therapy in PET+ disease? What would be the comparators?
Low sensitivity for detection of BMI in MCL is contrary to Hodgkins lymphoma and high grade B cell NHL where PET/CT may substitute BMB according to current guidelines. Please add the manuscript by F. Ricard et al. (J Nucl Med, 64 (1) (2023) pp 102-108).
Low sensitivity of PET for BMI was also demonstrated in follicular lymphoma. I suggest to add the manuscript by Y. Zheng, et al. (Ann Hematol, 102 (9) (2023), pp. 2403-2412)
Author Response
Thank you for giving us the opportunity to submit a revised draft of our manuscript titled “The Value of Bone Marrow Assessment by FDG PET/CT, Biopsy and Aspirate in the Upfront Evaluation of Mantle Cell Lymphoma: a Nationwide Cohort Study” to Cancers. We appreciate the time and effort that you and the reviewers have dedicated to providing your valuable feedback on our manuscript. We are grateful to the reviewers for their insightful comments on our paper.
We have been able to incorporate changes to reflect most of the suggestions provided by the reviewers. In addition, we provide changes to address the issues raised by the editorial team comments. Furthermore, we detected some minor errors in the manuscript draft which have now been corrected.
Here is a point-by-point response to either the reviewers’ comments, the concerns raised by the editorial team and the correction of minor errors.
Comments from Reviewer #1
Comment 1: This reviewer suggests to add information on patient selection for PET/CT. Was it performed in all patients with MCL? Add like a flow chart diagram to RESULTS to visualize recruitment, drop outs, missing data etc.
Response: Thank you for your valuable suggestion to include additional information on patient selection for PET/CT As outlined in the inclusion criteria (page 3, line 106), only patients with both BMO and PET performed at diagnosis were included in the study. A flow chart would indeed have been a very interesting addition to demonstrate how patients with MCL were selected for either test. However, unfortunately, we do not have access to that information, as the participating centers directly provided data only for those patients who underwent both assessments at diagnosis.
Comment 2: Please add to RESULTS: What where the criteria for PET positivity? There is merely a statement saying “visual analysis”. Did the investigators include any semiquantitative criteria like SUVs? How about reference organs like Liver, Bloodpool and normal Bone marrow uptake?
Response: We acknowledge the thoughtful issue raised by the reviewer. Image interpretation was made through qualitative (visual) analysis, considering the presence or absence of bone marrow involvement using glucose activity in the liver as a reference, which has now been specified (page 3, line 128). If present, bone marrow lesions were characterized as focal or diffuse, though the latter were discarded for this work. Data from the underlying CT scan and/ or patient’s medical history was also considered (stated in line 130). Thereafter, semi-quantitative analysis by the maximum standardized uptake value (SUVmax) normalized to body weight, measured the activity (voxel with maximum uptake) in a region or in the volume of interest was considered. Though a number of cut-off points have been considered, we herein set a cut-off of 9 to either describe the cohort and for the survival analysis (added to page 3, lines 132-135). Unfortunately, a number of SUVmax data were missed hampering the statistical analysis of this parameter as stated in the text. To further improve the comprehension of PET/CT methods and avoid unnecessary duplications, we now refer the reader to a detailed description in reference 11. Besides, as suggested, we have clarified the details regarding PET/CT BMI in the Results section (page 6, line 177).
Comment 3: This reviewer suggests to strengthen the DISCUSSION as follows: As long as knowledge of BMI is considered necessary for staging in MCL, PET/CT cannot substitute BMB. It should be pointed out, that based on the results reported, the clinical gain of PET/CT for MCL patients is not clear. Obviously, PET/CT identifies a subgroup of MCL patients with particularly aggressive and/or high volume BM disease. But could this be turned into a benefit for patients that justifies to perform this relatively expensive imaging test in all MCL patients? Specifically, could the authors propose a prospective, controlled study that evaluates intensified first-line therapy in PET+ disease? What would be the comparators?
Response: We appreciate the reviewer’s insightful comments, which have helped us refine the discussion and further clarify the implications of our findings (pages 13-14, lines 371-384). Regarding the suggestion to discuss the clinical benefit of PET/CT and its potential as a routine test for all mantle cell lymphoma (MCL) patients, we have expanded the discussion as follows:
- We have highlighted that, at present, identifying high-risk patients, whether BMI-PET/CT+ or based on previously described prognostic markers (e.g., TP53 mutations, Ki-67 expression), does not lead to therapeutic modifications in current clinical practice.
- We have included a summary of recent findings from the TRIANGLE trial, which evaluated the addition of ibrutinib to first-line treatment. This study demonstrated improved efficacy in younger MCL patients but with increased toxicity when combined with autologous stem cell transplantation. We have noted that the low number of patients treated with this regimen in our cohort limits the ability to draw specific conclusions.
- In alignment with the reviewer’s suggestion, we emphasize the importance of prospective, controlled studies to evaluate novel therapeutic strategies. Specifically, we propose that future studies should assess intensified or novel-agent-inclusive regimens for patients identified as PET-positive, especially those with high-risk MCL-PET-I scores. We suggest comparative trials that include intensified therapy versus current standard treatment protocols.
Comment 4: Low sensitivity for detection of BMI in MCL is contrary to Hodgkins’s lymphoma and high-grade B cell NHL where PET/CT may substitute BMB according to current guidelines. Please add the manuscript by F. Ricard et al. (J Nucl Med, 64 (1) (2023) pp 102-108).
Response: We thank the reviewer for this observation and for suggesting the manuscript by Ricard et al. We have added a discussion comparing our findings to those in high-grade B-cell NHL and Hodgkin’s lymphoma, where PET/CT has shown higher sensitivity for BMI detection. We have also included the suggested reference to support this comparison (page 12, lines 295-298).
Comment 5: Low sensitivity of PET for BMI was also demonstrated in follicular lymphoma. I suggest to add the manuscript by Y. Zheng, et al. (Ann Hematol, 102 (9) (2023), pp. 2403-2412)
Response: We appreciate the reviewer’s suggestion and have added the manuscript by Zheng et al. to highlight the similar low sensitivity of PET/CT for BMI in follicular lymphoma (page 12, lines 292-295). This reinforces our conclusion regarding the limitations of PET/CT in certain lymphoma subtypes.
Issues raised by the editorial team.
Comment 1. Please complete the information of table 3 header.
Response: Table 3 header has been modified as follows. “Diagnostic Performance of PET/CT and Bone Marrow Biopsy in Terms of Sensitivity, Negative Predictive Value and Accuracy for Detecting Bone Marrow Involvement.”.
Comment 2. Any research article describing a study involving humans should contain this statement.
Response: In the informed consent statement item, we now describe that Informed consent was obtained from all subjects involved in the study.
Comment 3. Please include the first ten authors' names before using “et al.” in the references.
Response: References have been modified according to your suggestions.
Correction of minor errors.
1 Line 80. “optional in the BMB evaluation” has been modified to “optional in the BM evaluation”
2.Lines 83 & 84 “either on stage (i.e., NCCN guidelines4, where BMI may be relevant)” has been been modified to “either on stage (i.e., NCCN guidelines4), where BMI may be relevant”.
3.Line 244 “PFS Model Score Calculation…” header. In line 3 of this paragraph: “ULN” has been modified to “UNL”.
Reviewer 2 Report
Comments and Suggestions for Authors
The authors of this study "The Value of Bone Marrow Assessment by FDG PET/CT, Biopsy and Aspirate in the Upfront Evaluation of Mantle Cell Lymphoma: a Nationwide Cohort Study", have effectively evaluated the prognostic impact of bone marrow infiltration (BMI) in mantle cell lymphoma (MCL) using PET/CT and bone marrow biopsy (BMB). They concluded that PET/CT provides greater prognostic value for progression-free survival (PFS) and overall survival (OS), particularly when combined with BMB, which remains essential for staging. The study introduces a new prognostic model, MCL-PET-I, that effectively stratifies patients into distinct risk groups, showing strong predictive value for clinical outcomes. The discussion is well-structured. The authors thoughtfully highlight the clinical significance of their findings, suggesting that incorporating PET/CT-based BMI into staging could improve prognostic accuracy. They also acknowledge the limitations of their work, such as the variability in PET/CT sensitivity and the study's reliance on a single cohort, which may affect the generalizability of their findings. Overall, the paper is comprehensive, and the conclusions are well-supported by the data, providing a meaningful contribution to the field of MCL research. Here are some minor suggestions.
· The study focuses mainly on PET/CT's role in detecting Bone Marrow Infiltration (BMI), but it could be useful to expand on how PET/CT's sensitivity in detecting extramedullary involvement might improve its staging accuracy. Since PET/CT might be more effective for identifying extramedullary disease, a more detailed examination of the combined impact of PET/CT on overall staging (considering both BMI and extramedullary disease) would provide more context for its clinical utility in staging lymphoma.
· Please mention in the manuscript if the sample size (n=148) provides adequate power to detect differences across all methods, especially for borderline non-significant results like BMI in PET/CT or BMB.
· In line 43, consider revising, “We deconstructed the IPI-NCCN, MIPI, and MIPI-c indices and considered BMI as either positive BMB, PET/CT, or a combination of both”, to “We deconstructed the IPI-NCCN, MIPI, and MIPI-c indices and considered BMI as positive if indicated by a BMB, PET/CT scan, or a combination of both”.
· Consider changing the color of the table headings for better visibility.
· The resolution of the Kaplan Meyer graphs could be improved.
· In line, 311, consider changing BM to BMI.
· Address whether BMI assessment techniques should be adapted based on patient characteristics, such as age, gender, or specific histological subtypes of MCL. For instance, could PET/CT be more relevant for younger or intensively treated patients?
· Include more details about the scoring system used in the MCL-PET-I index, such as the specific thresholds or weightings for parameters like LDH and age. This would make the index more immediately applicable for clinicians.
· Discuss the clinical implications of PET/CT's lower sensitivity and negative predictive value compared to BMB. For instance, how should clinicians approach cases with negative PET/CT results but high clinical suspicion of BMI?
Author Response
Thank you for giving us the opportunity to submit a revised draft of our manuscript titled “The Value of Bone Marrow Assessment by FDG PET/CT, Biopsy and Aspirate in the Upfront Evaluation of Mantle Cell Lymphoma: a Nationwide Cohort Study” to Cancers. We appreciate the time and effort that you and the reviewers have dedicated to providing your valuable feedback on our manuscript. We are grateful to the reviewers for their insightful comments on our paper.
We have been able to incorporate changes to reflect most of the suggestions provided by the reviewers. In addition, we provide changes to address the issues raised by the editorial team comments. Furthermore, we detected some minor errors in the manuscript draft which have now been corrected.
Here is a point-by-point response to either the reviewers’ comments, the concerns raised by the editorial team and the correction of minor errors.
Comments from Reviewer #2
Comment 1:The study focuses mainly on PET/CT's role in detecting Bone Marrow Infiltration (BMI), but it could be useful to expand on how PET/CT's sensitivity in detecting extramedullary involvement might improve its staging accuracy. Since PET/CT might be more effective for identifying extramedullary disease, a more detailed examination of the combined impact of PET/CT on overall staging (considering both BMI and extramedullary disease) would provide more context for its clinical utility in staging lymphoma.
Response: We appreciate the reviewer's suggestion to improve staging accuracy by combining the effect of PET/CT on the overall staging. We have statistically analyzed the data resulting from collecting BMI and extramedullary disease, but it was not overweighted in the final composite score, probably due to the limitations highlighted in the discussion (line 363-374).
Comment 2. Please mention in the manuscript if the sample size (n=148) provides adequate power to detect differences across all methods, especially for borderline non-significant results like BMI in PET/CT or BMB.
Response: We acknowledge this observation as strengthening the results obtained in our study. Considering the incidence of mantle cell lymphoma in the spanish population and taking into account some requirements for estimating the final sample (power, precision and confidence level), our study population is adequate, but given that this is a consecutive case study, we add this quote at beginning of the section “Method and patient” (page 3, lines 103-104).
Comment 3: In line 43, consider revising, “We deconstructed the IPI-NCCN, MIPI, and MIPI-c indices and considered BMI as either positive BMB, PET/CT, or a combination of both”, to “We deconstructed the IPI-NCCN, MIPI, and MIPI-c indices and considered BMI as positive if indicated by a BMB, PET/CT scan, or a combination of both”.
Response: We much appreciate the reviewer comment. It has now been corrected.
Comment 4: Consider changing the color of the table headings for better visibility.
Response: Again, we appreciate the reviewer comment. It has now been corrected.
Comment 5: The resolution of the Kaplan Meyer graphs could be improved.
Response: Again, we appreciate the reviewer comment. It has been improved.
Comment 6:In line, 311, consider changing BM to BMI.
Response: Again, we much appreciate the reviewer comment. It has now been corrected.
Comment 7: Address whether BMI assessment techniques should be adapted based on patient characteristics, such as age, gender, or specific histological subtypes of MCL. For instance, could PET/CT be more relevant for younger or intensively treated patients?
Response: We acknowledge the interesting point raised by the reviewer. We agree with him/her that it could be of interest to adapt not only BMI but much of the diagnostic effort to patients that will benefit from it. This was precisely the aim of analyzing separately the whole cohort and those patients intensively treated (supposedly “go-go” vs a mix of “go-go” and “no go”).
Comment 8: Include more details about the scoring system used in the MCL-PET-I index, such as the specific thresholds or weightings for parameters like LDH and age. This would make the index more immediately applicable for clinicians.
Response: We thanks the suggestion made by reviewer #2; we simplify the index and make it applicable for clinicians, we assign each parameter as 1 point after verifying the statistical significance using the Cox method and weighting their hazard ratio. To facilitate clinicians’ task the specific thresholds of LDH are according each local laboratory in a like manner of those used for the MIPI index; besides, to address age adjustment, the score was assigned after verifying the statistical significance to be dichotomized (added method and clarifications in page 8, lines 243-246).
Comment 9: Discuss the clinical implications of PET/CT’s lower sensitivity and negative predictive value compared to BMB. For instance, how should clinicians approach cases with negative PET/CT results but high clinical suspicion of BMI?
Response: We have incorporated your suggestion to discuss how to approach negative PET/CT results in cases with high clinical suspicion of BMI, to ensure accurate staging (Page 12, lines 302-305). The clinical implications are discussed in the following paragraphs, where we highlight that BMB is a much more sensitive technique for the detection of BMI, but PET/CT detected those cases that did worse in both PFS and OS, thus acting as a surrogate marker of aggressiveness.
Issues raised by the editorial team.
Comment 1. Please complete the information of table 3 header.
Response: Table 3 header has been modified as follows. “Diagnostic Performance of PET/CT and Bone Marrow Biopsy in Terms of Sensitivity, Negative Predictive Value and Accuracy for Detecting Bone Marrow Involvement.”.
Comment 2. Any research article describing a study involving humans should contain this statement.
Response: In the informed consent statement item, we now describe that Informed consent was obtained from all subjects involved in the study.
Comment 3. Please include the first ten authors' names before using “et al.” in the references.
Response: References have been modified according to your suggestions.
Correction of minor errors.
1 Line 80. “optional in the BMB evaluation” has been modified to “optional in the BM evaluation”
2.Lines 83 & 84 “either on stage (i.e., NCCN guidelines4, where BMI may be relevant)” has been been modified to “either on stage (i.e., NCCN guidelines4), where BMI may be relevant”.
3.Line 244 “PFS Model Score Calculation…” header. In line 3 of this paragraph: “ULN” has been modified to “UNL”.